# Effectiveness of a Real-Time X-ray Dosimetry Monitor in Reducing Radiation Exposure in Coronary Procedures: The ESPRESSO-Raysafe Randomized Trial

**DOI:** 10.3390/jcm10225350

**Published:** 2021-11-17

**Authors:** Maximilian Olschewski, Helen Ullrich, Moritz Brandt, Sebastian Steven, Majid Ahoopai, Recha Blessing, Aniela Petrescu, Philip Wenzel, Thomas Munzel, Tommaso Gori

**Affiliations:** Department of Cardiology, Cardiology 1, University Medical Center Mainz and Deutsches Zentrum für Herz und Kreislauf Forschung, Standort Rhein-Main, 55131 Mainz, Germany; maximilian.olschewski@unimedizin-mainz.de (M.O.); helen.ullrich@unimedizin-mainz.de (H.U.); moritz.brandt@unimedizin-mainz.de (M.B.); Sebastian.steven@unimedizin-mainz.de (S.S.); majid.ahoopai@unimedizin-mainz.de (M.A.); recha.blessing@unimedizin-mainz.de (R.B.); Aniela.petrescu@unimedizin-mainz.de (A.P.); wenzelp@uni-mainz.de (P.W.); tmuenzel@uni-mainz.de (T.M.)

**Keywords:** coronary artery disease, coronary stenting, radiation exposure

## Abstract

Background—Several methods to reduce radiation exposure in the setting of coronary procedures are available on the market, and we previously showed that additional radiation shields reduce operator exposure during radial interventions. We set out to examine the efficacy of real-time personal dosimetry monitoring in a real-world setting of radial artery catheterization. Methods and Results—In an all-comer prospective, parallel study, consecutive coronary diagnostic and intervention procedures were performed with the use of standard radiation shield alone (control group) or with the addition of a real-time dosimetry monitoring system (Raysafe, Billdal, Sweden, monitoring group). The primary outcome was the difference in exposure of the primary operator among groups. Additional endpoints included patient, nurse, second operator exposure and fluoroscopy time. A total of 700 procedures were included in the analysis (*n* = 369 in the monitoring group). There were no differences among groups in patients’ body mass index (*p* = 0.232), type of procedure (intervention vs. diagnostic, *p* = 0.172), and patient sex (*p* = 0.784). Fluoroscopy time was shorter in the monitoring group (5.6 (5.1–6.2) min vs. 7.0 (6.1–7.7) min, *p* = 0.023). Radiation exposure was significantly lower in the monitoring group for the patient (135 (115–151) µSv vs. 208 (176–245) µSv, *p* < 0.0001) but not for the first operator (9 (7–11) µSv vs. 10 (8–11), *p* = 0.70) and the assistant (2 (1–2) µSv vs. 2 (1–2) µSv, *p* = 0.121). Conclusions—In clinical daily practice, the use of a real-time dosimetry monitoring device reduces patient radiation exposure and fluoroscopy time without an effect on operator radiation exposure.

## 1. Introduction

Radiation exposure exposes patient and medical staff to dose-dependent damage (e.g., cataract for the operator and skin lesions for the patient), and it carries a stochastic risk for various severe diseases, including malignancies [1,2,3,4,5,6]. The risk of developing professional radiation-induced pathologies is directly proportional to the cumulative exposure to (scattered) radiations, as determined by the length of the career, the complexity and the type of procedures performed [7,8,9]. Awareness of this professional hazard is a cause of concern among operators, and according to a survey by the women’s initiative of the European Association for Percutaneous Coronary Interventions, exposure to radiation remains a barrier discouraging women from a career in interventional cardiology [10].

The exposure to radiation for patients and operators is determined by the total dose emitted by the X-ray apparatus (modern ones being more dose-sparing), but it can be modulated by the operator’s experience, the type of procedure performed, patient characteristics, and the use of protection devices [11,12,13]. Such devices are normally designed to reduce operator exposure and include, for instance, pads, screens, and aprons as well as robotic systems [14,15,16,17,18,19,20]. In an alternative approach, real-time dosimetry monitors provide the operators with instantaneous feedback with respect to the level of exposure, triggering virtuous behavior (e.g., increasing the distance between operator and X-ray source, decreasing the intensity of radiation, shortening fluoroscopy time, and reducing the use of radiation-intensive projections). Although such devices are on the market, evidence regarding their effectiveness in a real-life context remains limited.

## 2. Material and Methods

### 2.1. Study Design

The study was designed as prospective, single-centre, controlled, parallel group trial. Consecutive coronary procedures were performed using standard protection shields (control group, including a ceiling-mounted, 60 × 76 cm, 0.5 mm Pb, screen and overhanging 0.5 mm Pb panel curtain (Mavig, Munich, Germany)) or the same protection devices and a real-time X-ray dosimetry monitor (monitor group, Raysafe, Billdal, Sweden). Using this device, each staff is equipped with a personal dosimeter that records radiation exposure at 1-second intervals. Dosimetry data are presented real-time on a stand-alone display attached to the catheterization laboratory screen and placed next to the real-time fluoroscopy images. Color-coded bars (green, yellow, red) instantaneously indicate different levels of exposure for each operator, nurse and patient. This direct, easily accessible feedback allows taking immediate actions (e.g., increase distance from the radiation source, decrease the distance between patient and detector, reduce fluoroscopy time, etc.). In the monitoring group, this information was provided directly during the exam. In the control group, all dosimetry measurements were performed but the screen was darkened so that the information was only available for post hoc analysis. Staff in the catheterization was kept blinded at all times to the results of this analysis. All procedures performed in the catheterization laboratory equipped with the monitor were included in the analysis. The study was conducted between April and September 2020. The original plan to randomize periods of equal duration with and without monitor had to be abandoned because the coronavirus lockdown would have caused a significant imbalance between groups (only acute coronary syndromes, late presentations, and delays during lockdown). Therefore, we had to balance the length of “lockdown periods” between groups, and the initially planned randomization plan had to be abandoned. Sponsor of the study was the University Medical Center Mainz. The protocol was approved by the local ethics committee (reference number 2018-13051-KliFo). Since measurements did not introduce additional risks for the patients, the committee waived the need of an informed consent.

### 2.2. Hypothesis of the Study

We hypothesized that the use of radiation monitoring would be associated with a reduction in the operator’s radiation exposure in the setting of radial coronary diagnostic and interventional procedures.

### 2.3. Measurement of Scattered Radiation

Operators’ dosimetry was performed at chest height, outside the lead apron, with a personal dosimeter (Raysafe, Billdal, Sweden, Figure 1). The first and second operators and the nurse wore a personal dosimeter. An additional dosimeter was placed on the catheterization table to measure patient exposure. All procedural decisions were left to the operators’ discretion; the study procedures did not interfere with clinical routine except for the radiation protection device used.

Conventional radiation protection devices (lead aprons, lead collar, lead curtain and shield) were used in all procedures, which were performed using the same X-ray equipment (Philips AlluraClarity FD10, Philips Medical Systems, Eindhoven, The Netherlands).

### 2.4. Statistical Analysis

The primary endpoint of the study was the difference in exposure of the primary operator among groups. Secondary endpoint was the difference in relative exposure of the assistant operator, nurse, and patient, as well as fluoroscopy time and the difference in all outcomes in subanalyses limited to diagnostic versus interventional procedures, procedures in patients with BMI ≥30 vs. <30 or procedural duration >60 min. Data are presented as median [IQR] or *n* (%). Categorical data were analysed using the chi-square test; continuous variables were analysed using the Kruskal–Wallis test.

## 3. Results

### 3.1. Patient and Procedural Characteristics

The patients’ and procedural characteristics are presented in Table 1. Measurements were performed in a total of 700 procedures (369 in the monitoring group, 331 in the control group). There was a total of 347 (51%) coronary interventions (*p* = 0.172 among groups); 294 (42.8%) procedures involved also left ventricle angiography (*p* = 0.355), 33 (4.8%) right heart catheterization (*p* = 0.440) and 22 (3.2%) left ventricle biopsy (*p* = 0.654), in all cases without difference among randomization groups. Patients’ characteristics potentially associated with X-ray exposure risk (sex: *p* = 0.763, body mass index: *p* = 0.232) were also not different among groups. Finally, in the case of coronary interventions, parameters expressing procedural complexity (acute coronary syndromes, prevalence of chronic total occlusion procedures, use of additional imaging or hemodynamic tools, all *p* > 0.1) did not differ among groups. On average, 100 (61–151) and 92.5 (61–163) mL contrast medium was used per procedure (respectively, control and monitor group), without a difference between groups (*p* = 0.817).

### 3.2. Radiation Exposure

Mean fluoroscopy time was lower in the monitor group (control: 7.0 (6.1–7.7) min vs. 5.6 (5.1–6.2) min, *p* = 0.028). Mean dose-area product was also lower in the monitor group cGy·cm², *p* = 0.027). Exposure data are presented in Figure 2 and Table 2. The absolute exposure for both the first and the second operator was not different between groups (decrease in mean exposure for the first operator: 5.6%, *p* = 0.700; decrease in mean exposure for the assistant operator: 10.6%, *p* = 0.121). In contrast, nurse and patient exposure were lower in the monitor group (respectively, decrease in mean exposure for nurse: 18.5%, *p* = 0.012 and decrease in mean exposure for patients: 28.1%, *p* < 0.0001). When relative exposure (the ratio of absolute exposure to dosis/area product) was used, there was no difference between control and monitor group in first operator (*p* = 0.572), second operator (*p* = 0.755), and nurse exposure (*p* = 0.331). The relative exposure to the patient remained approximately 25% lower in the monitor group (*p* = 0.042).

Absolute exposure to staff and patients was expectedly higher in procedures longer than 60 min and in patients with a BMI > 30 (Table 3 and Table 4). There was no difference in the dose received by the operators and the nurse when only procedures with procedural time > 60 min were considered or patients with BMI > 30 (Figure 3 and Figure 4), but the difference in patient exposure was also maintained in these subanalyses. Interventional procedures (as compared to diagnostic ones) were not associated with additional radiation for staff or patients (Table 5).

## 4. Discussion

In a large-scale, all-comer, controlled study we tested the effect of a radiation dosimetry monitor with real-time feedback against the current standard (protection devices without monitor). We found that the use of the dosimetry monitor attenuated the absolute radiation exposure of the nurse and the patient (respectively, 18.5% and 28.1%) by reducing fluoroscopy time, while it only had a numerical, but statistically not significant, effect on operators’ dosis. These observations were maintained in subanalyses in interventional procedures and in procedures requiring prolonged fluoroscopy time, and they were independent of BMI.

### 4.1. Radiation Exposure in Interventional Cardiology and Interpretation of the Present Findings

Cardiac catheterization procedures account for about 40% of total medical radiation exposure [2,21]. Despite the progress in X-ray instrumentation, protection devices, the introduction of robotic PCI, and foremost the increased awareness, the progressively increasing complexity of the procedures performed (including structural and valvular procedures) leads to a constant increase in operator exposure [22,23,24]. This increase was recently quantified in a 54% higher average fluoroscopy time when procedures performed in the year 2016 were compared to those performed 10 years before [25]. This trend obviously represents an issue for new trainees; as well, according to a survey of the women’s initiative, radiation exposure may represent a barrier towards gender equality in this field [10].

In this scenario, a real-time dosimetry monitor was designed to trigger virtuous behavior by providing immediate color-coded feedback to the operators. The warning signals on the monitor may indeed foster the use of shields, reduce the use of projections (such as the left caudal), reduce the fluoroscopy time and increase the distance between X-ray source and persons exposed. Obviously, not all of these interventions reduce equally the exposure of operators, nurses, and patients. For instance, by raising the table to reduce the distance between patient and X-ray detector, one would be expected to reduce scattered radiation (and thus primarily staff exposure). A similar effect would be expected following increased use of protective shields. By reducing fluoroscopy time, one would in contrast reduce the exposure of both staff and patient. In a recent report, the group of Schulz et al. reported a reduction in absolute radiation exposure by approximately 60% following introduction of the Raysafe monitor without significant changes in fluoroscopy time, dose-area product, or patient characteristics [26]. In our study, despite a ~2.5 times larger sample size, the ~10–15% reduction in operator exposure (for both main and assistant operator) did not reach statistical significance. In contrast, the use of the real-time monitor led to a significant reduction in fluoroscopy time and, therefore, in the exposure to patient and nurses. The reason for this difference in the two studies can only be hypothesized. Since the impact of any additional protection device varies depending on the context of the laboratory where it is used, the local availability of shielding devices and the motivation of team members to react to the information provided by the monitor play an important role. In the present experience, the use of the monitor triggered a reaction (reduction in fluoroscopy time leading to a reduction in exposure of patients and nurses). In contrast, the effect on operators, protected by the devices described in our previous paper, was minimal [27].

### 4.2. Limitations

Our study has several limitations. First, the number of included patients was limited and it is possible that statistical significance might have been reached with a larger sample. However, the current sample size was similar to that of our previous study where we showed that physical X-ray protection devices have a quantitatively much larger effect in reducing exposure [20]. The use of these devices is therefore more relevant towards reducing exposure. It is possible that, over a period of several years, milder reductions (in the range of 5–10% for the first and second operator) might also lead to a reduction in radiation-induced professional diseases. All procedural steps such as access route, use of projections, and use of protection devices were left to the operator’s discretion in order to reproduce routine practice. It might be that Raysafe is more efficient in some than other settings.

As expected, we observed a large variability in radiation exposure and in fluoroscopy use; further analysis of smaller subgroups is therefore complicated. In line with guidelines, radiation exposure was measured at one single level for each staff member and the patient.

Finally, this was a single-center study and it is possible that, as described above, the effect of real-time monitoring would be different in different settings.

## Figures and Tables

**Figure 1 jcm-10-05350-f001:**
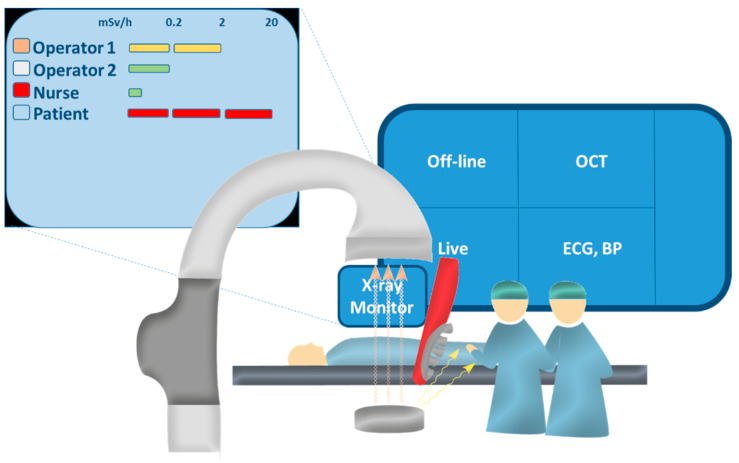
Real-time measurement of dosimetry.

**Figure 2 jcm-10-05350-f002:**
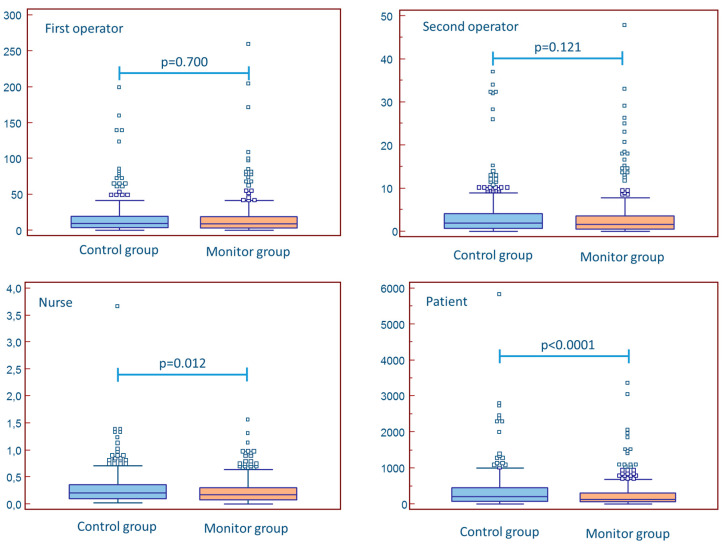
The impact of real-time monitoring on operator, nurse, and patient exposure.

**Figure 3 jcm-10-05350-f003:**
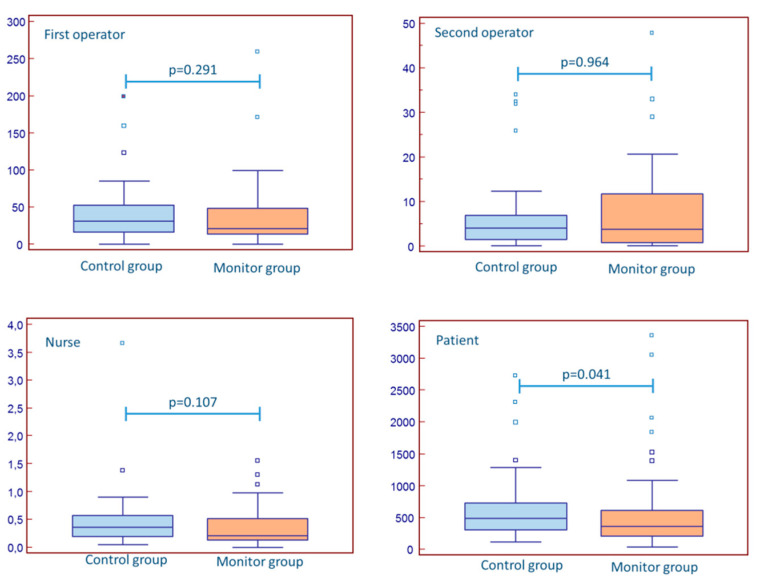
The impact of real-time monitoring in procedures >60 min.

**Figure 4 jcm-10-05350-f004:**
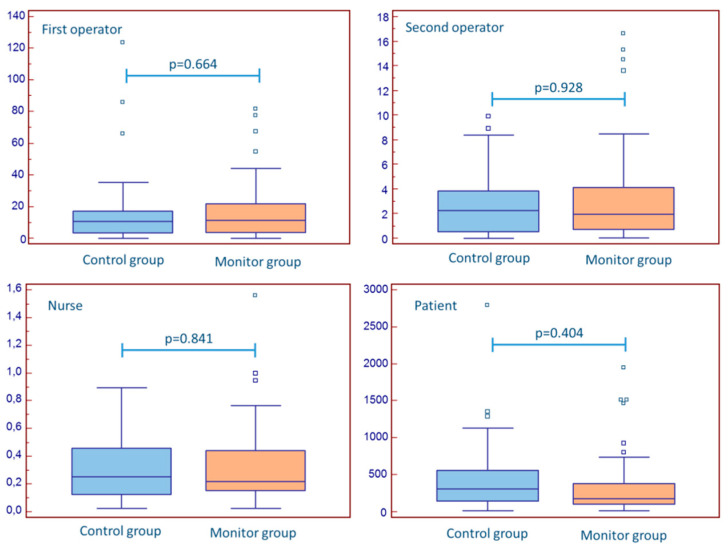
The impact of real-time monitoring in procedures in patients with BMI > 30.

**Table 1 jcm-10-05350-t001:** Procedure characteristics.

	Control	Monitor	All Procedures	*p*
	*n* (%)/Median [IQR]	*n* (%)/Median [IQR]	*n* (%)/Median [IQR]	
Procedures, total	331 (47%)	369 (53%)	700 (100%)	
PCI	176 (53.2%)	175 (47.4%)	347 (51.1%)	0.172
LV Angiography	146 (44.1%)	148 (40.1%)	294 (42.8%)	0.355
FFR/iFR	27 (8.2%)	19 (5.1%)	46 (6.7%)	0.153
OCT	9 (2.7%)	9 (2.4%)	18 (2.6%)	0.984
Recanalization of chronic total occlusions	3 (0.9%)	3 (0.8%)	6 (0.9%)	0.776
Rotablation	1 (0.3%)	5 (1.4%)	6 (0.9%)	0.269
Right heart catheterization	13 (3.9%)	20 (5.4%)	33 (4.8%)	0.440
Biopsy	12 (3.6%)	10 (2.7%)	22 (3.2%)	0.654
Male Patient	220 (66.5%)	239 (64.8%)	459 (65.6%)	0.763
Unstable angina	58 (17.5%)	56 (15.2%)	114 (16.6%)	0.485
NSTEMI	51 (15.4%)	66 (17.9%)	127 (17.0%)	0.414
STEMI	47 (14.2%)	41 (11.1%)	88 (12.8%)	0.278
Hyperlipoproteinemia	177 (53.5%)	197 (53.4%)	374 (54.4%)	0.997
Dialyse	8 (2.4%)	14 (3.8%)	22 (3.2%)	0.400
BMI [Kg/m²]	26.6 (24.3–29.4)	26.1 (24.2–28.3)	26.2 (24.2–29.1)	0.232
Access radial left	20 (6.0%)	19 (5.1%)	39 (5.7%)	0.743

Patient characteristics. BMI: body mass index. CTO: chronic total occlusion. FFR: fractional flow reserve. iFR: instantaneous flow reserve. LV: left ventricle. OCT: optical coherence tomography. PCI: percutaneous coronary intervention.

**Table 2 jcm-10-05350-t002:** The impact of real-time dosimetry monitoring.

	Group			
	Control	Monitor	All Procedures	*p*
	Median [IQR]	Median [IQR]	Median [IQR]	
Contrast medium [mL]	100 (61–151)	92.5 (61–163)	93 (61–156)	0.817
Fluoroscopy time [min]	7.0 (6.1–7.7)	5.6 (5.1–6.2)	6.1 (3.2–10.7)	0.028
Dose-Area product, cGy·cm^2^	21.8 (12.5–40.2)	18.8 (16.2–21.6)	20.0 (11.4–36.4)	0.027
Dosimeter (E, first operator) [µSv]	9.9 (3.8–19.9)	9.0 (3.5–19.2)	9.2 (3.5–19.7)	0.700
Dosimeter (E, assistant operator) [µSv]	1.9 (0.7–4.1)	1.6 (0.6–3.6)	1.7 (0.6–3.8)	0.121
Dosimeter (E, nurse) [µSv]	0.20 (0.10–0.36)	0.17 (0.08–0.31)	0.18 (0.09–0.33)	0.012
Dosimeter (E, patient) [µSv]	208 (87–455)	135 (68–318)	162 (74–379)	<0.0001

**Table 3 jcm-10-05350-t003:** Exposure in procedures lasting >60 min.

	Group			
	Control	Monitor	All Procedures	*p*
	Median [IQR]	Median [IQR]	Median [IQR]	
Contrast medium [mL]	104 (59–154)	91 (62–186)	100 (60–169)	0.974
Fluoroscopy time [min]	18.0 (13.7–23.6)	19.4 (14.3–28.3)	18.3 (13.9–24.4)	0.755
Dose-Area product, cGy·cm^2^	24.3 (13.0–41.0)	17.7 (11.8–36.6)	20.6 (12.7–37.5)	0.291
Dosimeter (E, first operator) [µSv]	31.0 (16.8–52.7)	21.4 (14.1–48.4)	23.4 (16.0–51.1)	0.291
Dosimeter (E, assistant operator) [µSv]	4.1 (1.5–6.9)	3.84 (0.9–11.8)	4.1 (1.0–7.6)	0.964
Dosimeter (E, nurse) [µSv]	0.36 (0.20–0.57)	0.20 (0.13–0.51)	0.29 (0.15–0.56)	0.107
Dosimeter (E, patient) [µSv]	489 (310–733)	369 (215–618)	420 (243–656)	0.041

**Table 4 jcm-10-05350-t004:** Exposure in procedures in patients with BMI > 30.

	Group			
	Control	Monitor	All Procedures	*p*
	Median [IQR]	Median [IQR]	Median [IQR]	
Contrast medium [mL]	90.5 (60.0–137.0)	89.5 (60.0–130.0)	90.0 (60.0–132.5)	0.877
Fluoroscopy time [min]	7.6 (3.5–11.1)	5.6 (3.8–9.1)	6.0 (3.6–10.2)	0.345
Dose-Area product, cGy·cm^2^	18.9 (11.7–33.1)	16.4 (13.1–21.5)	16.9 (11.0–30.0)	0.401
Dosimeter (E, first operator) [µSv]	10.6 (3.6–17.2)	11.5 (3.7–22.0)	11.1 (3.7–20.6)	0.665
Dosimeter (E, assistant operator) [µSv]	2.3 (0.6–3.8)	2.0 (0.7–4.1)	2.1 (0.6–3.9)	0.928
Dosimeter (E, nurse) [µSv]	0.25 (0.13–0.46)	0.22 (0.15–0.44)	0.23 (0.15–0.45)	0.841
Dosimeter (E, patient) [µSv]	315.0 (147.2–555.3)	184.3 (110.9–386.4)	230.2 (115.8–468.1)	0.044

**Table 5 jcm-10-05350-t005:** Exposure in interventional procedures.

	Group			
	Control	Monitor	All Procedures	*p*
	Median [IQR]	Median [IQR]	Median [IQR]	
Contrast medium [mL]	143.0 (103.0–153.0)	147.0 (100.3–192.8)	143.5 (102.0–193.0)	0.859
Fluoroscopy time [min]	6.8 (3.5–11.1)	5.6 (3.2–9.3)	6.0 (3.3–10.6)	0.054
Dose-Area product, cGy·cm^2^	33.3 (22.2–49.2)	32.0 (20.0–45.0)	32.4 (21.4–46.7)	0.196
Dosimeter (E, first operator) [µSv]	10.0 (3.8–21.5)	9.5 (3.3–19.2)	9.9 (3.5–20.1)	0.756
Dosimeter (E, assistant operator) [µSv]	1.9 (0.7–4.0)	1.6 (0.5–4.2)	1.7 (0.6–4.1)	0.237
Dosimeter (E, nurse) [µSv]	0.20 (0.10–0.44)	0.18 (0.08–0.33)	0.19 (0.10–0.36)	0.138
Dosimeter (E, patient) [µSv]	222.3 (95.6–468.7)	126.5 (68.4–305.8)	170.8 (76.5–398.8)	0.001

## Data Availability

Data are available upon motivated request.

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
