# Peer review of "Effectiveness of a Real-Time X-ray Dosimetry Monitor in Reducing Radiation Exposure in Coronary Procedures: The ESPRESSO-Raysafe Randomized Trial"

_jcm, 2021, doi:10.3390/jcm10225350_

Round 1

Reviewer 1 Report

General comments

This is a single-center prospective and controlled parallel-group trial that aims to investigate the benefits of a real-time X-ray dosimetry monitor in reducing operator´s radiation in the setting of radial coronary diagnostic and interventional procedures. The study included 700 procedures (369 in the monitoring and 331 in the control group).

The main objective was not achieved, no differences were found in operator’s radiation between groups, although there was a reduction in fluoroscopy time and patient’s radiation in the monitoring group (the group with real-time X-ray dosimetry).

Although this study had several limitations such as single-center with a limited number of patients included or no information about long-time follow-up operator’s radiation, the reduction in fluoroscopy time is very interesting.

The manuscript is clear, well written, the tables simple but well defined. The manuscript has an interesting and simple message: making conscious of the radiation with real-time measures could modify the operator’s techniques with an important reduction in fluoroscopy time and patient’s radiation.

Author Response

Your comment:

[…] The manuscript is clear, well written, the tables simple but well defined. The manuscript has an interesting and simple message: making conscious of the radiation with real-time measures could modify the operator’s techniques with an important reduction in fluoroscopy time and patient’s radiation.

Our response: We thank you very much for your positive comments, we are glad that you appreciate our work.

Reviewer 2 Report

The manuscript entitled “Effectiveness of a real-time x-ray dosimetry monitor in reducing radiation exposure in coronary procedures: the ESPRESSO-Raysafe randomized trial” has been reviewed with a great interest. The authors demonstrated significantly reduced radiation exposure to the patients with real-time dosimetry monitors on the patients and operators. There are some points to be addressed.

Major points

1) Was this a randomized study? If it is the case, the authors should clearly state that in the section of study design.

2) Although the authors stated that this is a parallel study, the number of enrolled patients were different between the two groups. The authors should show the study flow for the patient recruitment, enrollment, and exclusion.

3) Page 3 Line 99: “Conventional radiation protection measures were used in all procedures.” What were the correlations between real-time dosimetry and conventional dosimetry?

4) An illustrated information of real-time x-ray dosimetry monitor would be helpful to understand the system.

Minor point

1) Page 4 Line 118: “a left ventricle biopsy”: “a” should be removed.

2) Table 1: Total number of hyperlipoproteinemia is missing.

3) Page 8 Line 179: “BMI larger or smaller than 30”  Does this mean everyone?

Author Response

Your comment: The manuscript entitled [..] has been reviewed with a great interest. The authors demonstrated  […]. There are some points to be addressed.

 Ozr response: Thank you very much for your interest and your help in improving the manuscript. We hope you will find our changes, as detailed below, satisfactory.

Your comment: 1) Was this a randomized study? If it is the case, the authors should clearly state that in the section of study design.

Our response: Your comment is well taken, the study was unfortunately not randomized for logistical reasons. In order to make the Raysafe monitor as visible as possible, it needs to be installed on the cath lab monitor, directly in front of the operators (see picture), which requires some work and cannot be modified on a single-patient basis. In order to mix our cases as much as possible, we had initially planned to randomize periods of one-two weeks with the monitor with periods without the monitor. Unfortunately, we later realized that we had to balance not only holiday periods (where the prevalence of acute coronary syndromes is higher) between the two groups, but also (much less predictable) periods of corona virus-lockdown, when elective procedures were also limited. This meant that we had to balance the length of „lockdown periods“ between groups in order to avoid unbalances between groups; the initially planned randomization plan had to be abandoned. At the end, the cohorts were recruited during periods of equal length, but due to corona it was impossible to stick to the randomized plan because this would have caused significant unbalances between groups. Practically, the order of the periods was decided by factors independent of the study (and each „group“ had an equal chance to recruit in a given moment). However, formally, the duration and length of the periods was not fixed before beginning of the study and in all honesty we cannot state this.

This is now described in lines 85-91 (methods section).

2) Although the authors stated that this is a parallel study, the number of enrolled patients were different between the two groups. The authors should show the study flow for the patient recruitment, enrollment, and exclusion.

Thank you very much for this question. This is partially answered above: the periods were equal in duration, and holidays and lockdown periods were balanced between groups. No procedure occurring during these periods was excluded. The difference (~10%) in th enumber of procedures performed in th two groups is random.

3) Page 3 Line 99: “Conventional radiation protection measures were used in all procedures.” What were the correlations between real-time dosimetry and conventional dosimetry?

By „conventional radiation protection measures“ we meant lead aprons, shields, and a lead curtain. We now exchanged „measures“ with devices“ to avoid this confusion. Standard dosimetry is (by law) performed in our laboratory with a single dosimeter placed below the lead apron; the results are almost invariably below the minimum threshold, so it is impossible to use these data.

Your comment: 4) An illustrated information of real-time x-ray dosimetry monitor would be helpful to understand the system.

Our response: Thank you very much for this idea, we have added a new figure 1 to describe the scenario.

Minor point

1) Page 4 Line 118: “a left ventricle biopsy”: “a” should be removed.

2) Table 1: Total number of hyperlipoproteinemia is missing.

3) Page 8 Line 179: “BMI larger or smaller than 30”  Does this mean everyone?

Our response: Thank you very much, all these mistakes have been edited.